# Effectiveness of a Nurse-Led Telecare Programme in the Postoperative Follow-Up of Bariatric Surgery Patients: A Quasi-Experimental Study

**DOI:** 10.3390/healthcare12232448

**Published:** 2024-12-05

**Authors:** María de los Ángeles Maqueda-Martínez, Manuel Ferrer-Márquez, Manuel García-Redondo, Francisco Rubio-Gil, Ángel Reina-Duarte, José Granero-Molina, Matías Correa-Casado, Anabel Chica-Pérez

**Affiliations:** 1Bariatric Surgery Department, Servicio de Cirugía General y Digestiva, Hospital Universitario Torrecárdenas, 04009 Almería, Spain; 2Department of Nursing, Physiotherapy and Medicine, University of Almeria, 04120 Almería, Spainmcc249@ual.es (M.C.-C.);; 3Facultad de Ciencias de la Salud, Universidad Autónoma de Chile, Santiago 7500000, Chile

**Keywords:** bariatric surgery, nurse-led clinic, telemedicine, postoperative care

## Abstract

Background/Objectives: Obesity is a growing public health challenge due to its high prevalence and associated comorbidities. Bariatric surgery is the most effective treatment for achieving sustained weight reduction when more conservative treatments have failed. This study evaluates the impact of a nurse-led telecare follow-up programme in the immediate postoperative period for patients who have undergone bariatric surgery. Methods: A quasi-experimental study was carried out in two hospitals in southern Spain. We included 161 patients who met the inclusion criteria: a body mass index (BMI) ≥ 40 kg/m^2^ or a BMI ≥ 35 kg/m^2^ with associated comorbidities, and the failure of non-surgical treatments. Patients were divided into two groups: the intervention group (IG), which received follow-up telephone calls from a specialised nurse during the first 30 days post-surgery, and the control group (CG), which received standard care. The nurse, who was available 24 h a day, answered questions and dealt with queries over the phone or referred patients to the emergency department if necessary. Several variables were recorded, including the number of telephone consultations, reasons for consultation, number of emergency visits, readmissions, and surgical reinterventions. Results and Conclusions: The IG showed a significant reduction in ED visits (4.9% vs. 30% in CG), and consultations were mainly related to diet and drainage. The nurse telecare intervention significantly improved postoperative recovery by reducing complications and optimising the safety and quality of postoperative care. These results reinforce the importance of personalised follow-up in improving clinical outcomes in bariatric patients.

## 1. Introduction

Obesity is a major public health issue in all developed countries due to its growing prevalence, the increased risk of morbidity and mortality from associated medical complications, and the related healthcare costs [1]. The main way of treating obesity is based on prevention [2,3,4]. Thus, holistic interventions to promote healthy lifestyles are the first step in the clinical approach to obesity [2,3] and are the basis of both the prevention and clinical management of the condition. When these measures fail, the most effective option in terms of adequate and sustained weight loss is surgery [2,4,5]. In 2022, the American Society for Metabolic and Bariatric Surgery (ASMBS) and the International Federation for the Surgery of Obesity and Metabolic Disorders (IFSO) published a new document updating the surgical recommendations for patients with a body mass index (BMI) ≥ 35 kg/m^2^ or patients with a BMI between 30 and 34.6 kg/m^2^ who have a metabolic disease [6].

In functional terms, bariatric surgery techniques can be divided into purely restrictive, mixed (with a restrictive and hypoabsorptive component), and mixed with a predominantly malabsorptive component [7,8,9]. Laparoscopic vertical gastrectomy [10,11], laparoscopic gastric bypass [12,13], and SADI-S (single-anastomosis duodeno-ileal bypass with sleeve gastrectomy) [14] are currently the main examples in each group.

Hospital discharge in patients undergoing bariatric surgery requires rather complex care [15,16,17,18,19]. In these circumstances, nurses play a key role in promoting the overall quality of care by improving patient safety, wellbeing, and stress levels [18,19,20]. In patients who have undergone bariatric surgery, the first month post-intervention is associated with certain adverse health outcomes [21], including an impaired ability to perform basic activities of daily living [22], the loss of muscle mass, malnutrition, sedentary behaviour, and the development of haemorrhoids and surgical wound infections [23]. In addition, patients who undergo bariatric surgery experience stress and fear and are unaware of how to respond to certain circumstances that arise during this time [24,25,26,27]. In fact, evidence suggests that people who have undergone bariatric surgery tend to visit emergency departments more frequently and have more hospital admissions during the first month post-surgery [28]. In light of this, nurses should work with the rest of the interdisciplinary team to design, implement, and evaluate interventions that promote the health and safety of bariatric surgery patients during the first month following the intervention [18,19].

Previous research suggests that nutritional and psychological behavioural support interventions may contribute to positive outcomes in patients who have undergone bariatric surgery [17,26,29]. Preoperative interventions that include exercise programmes and training on nutrition and lifestyle changes have been shown to improve adherence to postoperative recommendations [30,31,32]. Intensive postoperative follow-up interventions have also been implemented and evaluated and have been shown to significantly improve patients’ functional recovery and quality of life after bariatric surgery [30,31,32]. In addition, an interdisciplinary approach to care in patients following bariatric surgery is known to improve outcomes and reduce long-term complications [18,19]. However, there is a lack of research on the effects of nurse-led interdisciplinary interventions focusing on personalised telecare in the first postoperative month for patients who have undergone bariatric surgery.

Telecare can be defined as a tool by which healthcare professionals use communications technology to provide health and social care directly to patients [33,34]. From a theoretical point of view, telecare interventions have the potential to improve patient outcomes, providing they are developed and implemented by competent healthcare professionals who are aware of the patient’s context and work collaboratively in interdisciplinary teams to foster the engagement of patients in the decision-making process [34]. In this regard, the available evidence suggests that multidisciplinary telecare interventions could improve both clinical and patient-reported outcomes in surgical patients [35]. In fact, nurse-led multidisciplinary telecare interventions could effectively improve patients’ satisfaction, foster adherence to postoperative protocols, reduce postoperative complications, and enhance recovery through better coordination of care and timely follow-up [36,37,38].

## 2. Materials and Methods

### 2.1. Objectives and Hypothesis

The objectives of our study were (1) to describe the nature of consultations in patients who have undergone bariatric surgery in the immediate postoperative period after discharge (first month post-surgery) and (2) to analyse the effects of the intervention (the implementation of a nurse-led telecare post-surgical follow-up programme) on complications and readmissions in these patients.

The hypothesis of this study was that a nurse-led telecare follow-up programme would reduce complications and optimise the safety and quality of immediate post-surgical care in people who had undergone bariatric surgery.

### 2.2. Design

A quasi-experimental study was designed and conducted in two hospitals in southern Spain. This manuscript was written following the recommendations of the Transparent Reporting of Evaluations with Nonrandomized Designs (TREND) guidelines for reporting quasi-experimental studies [39].

### 2.3. Setting and Participants

This study included patients who had undergone bariatric surgery and been examined by a multidisciplinary team (an endocrinologist, nutritionist, psychologist, nurse, and surgeon). The surgical techniques performed were single-anastomosis gastric bypass, laparoscopic vertical gastrectomy, or simplified gastric bypass, depending on the characteristics of each patient and the established department protocol. The inclusion criteria were as follows: (1) a BMI ≥ 40 kg/m^2^ or a BMI ≥ 35 kg/m^2^ with associated comorbidities, (2) patient aged between 18 and 65 years, (3) obesity persisting for more than 5 years, and (4) the failure of other non-surgical treatments to treat obesity. Patients were excluded from the study if they had any physical or psychological condition that prevented them from having a telephone conversation and/or lacked the means of communication (mobile technology) to carry out the intervention (teleconsultation).

### 2.4. Sample Size Calculation

The sample size was calculated on the basis of the main study variable, ‘number of visits to the emergency department’, using G-Power software 3.1.9.6 for Mac (Heinrich-Heine-University Düsseldorf, Düsseldorf, Germany). At the time when the study was designed, no evidence had been found on the specific effects of nurse-led telecare interventions on the number of ED visits in the first 30 days after the surgery. However, it was known that around 17.5% of patients who had undergone bariatric surgery visited the emergency department in the first 30 days after surgery [40]. Additionally, it was also apparent that other programmes incorporating a post-surgery call from healthcare staff reduced ED visits by 54% in the first 90 days [41]. The hypothesis of this study was that a programme led by a surgical nurse answering ad hoc calls from patients could improve the number of ED visits by 50% in the first 30 days. Therefore, based on an alpha error of 0.05, 77 patients per group would be needed to achieve a statistical power of 80%.

### 2.5. Recruitment

Recruitment was carried out over 18 months (January 2019 to June 2020) using a non-probability consecutive sampling technique. In this quasi-experimental study, patients were not randomly allocated to the intervention or the control group, as one of the two participating hospitals could not offer the nurse-led telecare intervention. Therefore, patients from the institution that offered the nurse-led telecare intervention were allocated to the intervention group and all patients from the other institution were allocated to the control group. The nurse, who was specialised in bariatric surgery (certified by courses endorsed by SECO—Spanish Society of Bariatric Surgery) and a member of the multidisciplinary team for bariatric surgery, was in charge of collecting all the data related to the study. This nurse met the patients in the consultation room together with the surgeon and explained the procedure to be followed. The data of the patients in the intervention group (IG) were collected from the patients’ medical records and subsequently stored in a database created specifically for the study. The results of telephone follow-ups in the postoperative period (at least 30 days after surgery and following discharge from hospital) were added directly to the database. The same nurse collected the data of the control group (CG) from the patients’ medical records. Recruitment was carried out sequentially (one by one) as patients underwent surgery and entered their postoperative period.

### 2.6. Intervention

All patients (both CG and IG) who met the criteria were discharged 48 h after surgery and were put on a liquid diet and nutritional supplements.

Control Group (CG): The Control Group were given a detailed explanation (both orally and in writing) of the care that they would require during the postoperative period (the usual care from the Andalusian Public Healthcare System, hereafter referred to as APHS) until the following consultation. This included information about surgical wound management and drainage (in the cases of patients discharged with drains), the type of diet that they should follow (a liquid diet and nutritional supplements for the first 7 days after discharge, followed by a semi-liquid diet for the next three weeks), the treatment to be followed (explaining how to administer the low-molecular-weight heparin), and the type of exercise that they could and should perform (walking daily for one hour after hospital discharge). They were given an appointment for their next check-up at the surgery clinic, as well as an appointment at the nutrition clinic 30 days post-surgery. They were informed that, if they had any problems or queries, they could go to their primary care doctor or to the emergency department (in the event of fever or abdominal pain that did not subside with their usual analgesia, incoercible vomiting, or changes in the appearance of the drainage). All of the above was provided with a written protocol at the time of discharge from hospital.Intervention Group (IG): All patients received the same information after discharge (usual APHS care) as the CG. In addition, they were informed that, if they were to have any queries or incidents, they could message (via WhatsApp) or call the specialist nurse, who was available 24 h a day, 7 days a week.

All calls related to postoperative care, as well as dietary queries, could be addressed directly by the nurse over the telephone. Calls requiring the assessment of the patient by a doctor (fever, abdominal pain that did not subside with the usual analgesia, incoercible vomiting, or changes in the appearance of the drainage) were dealt with by referring the patient to the emergency department.

### 2.7. Study Variables

The variables used to assess the patients’ characteristics were as follows:Body weight. This was measured in kilogrammes (kg) and obtained after weighing the patient on the scale in the consulting room at the first preoperative visit and 30 days post-surgery.Body mass index (BMI). This was obtained by applying the kg/m^2^ formula.Existence of comorbidities. These included the main conditions associated with bariatric patients, as well as existing health conditions in their clinical history (arterial hypertension, diabetes mellitus, sleep apnoea syndrome, or others).Type of surgical intervention. Depending on age, BMI, and associated health conditions, the intervention to be performed was determined (laparoscopic vertical gastrectomy, single-anastomosis gastric bypass, or simplified gastric bypass).Intraoperative complications. This included the presence of intraoperative bleeding requiring transfusion, or organ injury (spleen, liver, small intestine, or large intestine).Postoperative complications. These were classified according to Clavien Dindo [42].Days spent in hospital. The number of days spent in hospital from the operation to hospital discharge was calculated.


The outcome variables used to assess the effects of the intervention were as follows:
Number of telephone consultations (calls/WhatsApp). Telephone consultations were numbered and divided into calls or written consultations (WhatsApp).Time of consultation: The times at which patients contacted the nurse by telephone were recorded and divided into morning shift (7–15 h), afternoon/evening shift (15–23 h), and night shift (23–7 h).Reason for telephone consultations. The reasons for consultations were divided into dietary questions (queries about the protocol given to the patient on discharge), problems with drainage, fever (over 38 °C), abdominal pain, vomiting, problems with surgical wounds, intestinal transit disorders (constipation or diarrhoea), pyrosis, dizziness, and other consultations.How the telephone consultations were resolved. Depending on the nature of the queries or problems raised by the patient over the telephone, there were two options: a solution provided by the nurse over the telephone or referral to the emergency department.Visits to the emergency department. This included patients who, either on their own account or after speaking to the specialist nurse over the phone, went to the emergency department with a problem after medical discharge and up to 30 days after the surgical intervention.Hospital readmissions. Patients who were admitted to the surgery department after being assessed in the emergency department.Surgical re-interventions. Patients who required urgent surgical intervention after hospital discharge (within 30 days after the first intervention) due to complications related to the bariatric surgery performed.Days in hospital after readmission. The length of stay from readmission to hospital discharge was recorded.

### 2.8. Data Collection Procedure

The data collection process was carried out over 18 months. All patients (both CG and IG) were given appointments 30 days after surgery. During this consultation, their progress was assessed by the surgeon and the specialist nurse, who recorded it in their clinical history (from which the variables to be studied were taken). The nurse added the telephone consultations (calls or via WhatsApp) made by the IG patients to the database, and an external senior researcher double-checked the data.

### 2.9. Data Analysis

The data analysis was performed using SPSS v.29 (SPSS Inc., Chicago, IL, USA). To address the first aim of the study, descriptive statistics in terms of means, frequencies, and percentages were computed to describe the sample’s sociodemographic characteristics and the participants’ scores in the main outcome variables. In order to analyse the effects of the intervention on complications and readmissions in patients who had undergone bariatric surgery, bivariate analyses were performed to compare the two groups (control and experimental) for the following outcome variables: ED visits, hospital readmissions, surgical re-intervention, and the mean length of readmission stay (in days). Between-groups differences were tested using the Mann–Whitney U-test.

### 2.10. Ethical Considerations

The study was approved by Almería’s County Ethics Committee for Biomedical Research (protocol code 9/2019 and 22/01/2019). The patients invited to participate were informed about the aim of the study and of the voluntary nature of their participation. Confidentiality and anonymity were respected in accordance with the ethical principles of the Declaration of Helsinki. All participants signed an informed consent form before starting the data collection questionnaires and after reading a document with information about the study.

## 3. Results

The number of participants included in the study was 161, divided into the CG (*n* = 80) and the IG (*n* = 81). The characteristics of the population are shown in Table 1. Regarding gender distribution, a total of 106 patients (65.8%) were female, with a mean age of 44.21 ± 9.65 years. The BMI of the participants was 44.52 ± 7.35 kg/m^2^. The main comorbidities in the sample were hypertension (HT) in 38.5% of patients, diabetes in 16.8% of patients, and obstructive sleep apnoea syndrome (OSAS) in 37.9% of cases.

Regarding the surgical procedures performed, single-anastomosis gastric bypass (SAGB) was performed on a total of 108 (67.1%) patients, laparoscopic vertical gastrectomy was performed on 24 (14.9%), and simplified gastric bypass was performed on 23 (14.3%) patients. There were no intraoperative complications during the surgical procedures.

Pre-discharge complications (5.5%) were classified as grade I according to the Clavien–Dindo classification [35]. There were no reoperations and no deaths. The mean length of hospital stay was 2.94 ± 6.01 days.

In terms of the timing of the telephone consultations made in the IG after discharge (Table 2), we observed that most of them (54%) took place in the morning shift (7:00–15:00), 42.02% took place during the afternoon/evening shift (15:00–23:00), and a minority (3.64%) occurred during the night shift (23:00–07:00). The mean number of consultations per patient was 4.26 ± 3.98, of which more were made via WhatsApp (65.3%) than by telephone (34.7%). In 96.1% of the cases, the issue was resolved directly over the phone. Only four patients (4.9%) were referred to the emergency department and required hospital admission.

The main reasons why IG patients made contact after discharge were (Table 3) dietary concerns (26.8%), drainage problems (11.5%), surgical wound problems (5.8%), constipation (4.9%), abdominal pain (4.9%), and vomiting (3.7%), all of which were directly related to the surgery.

When comparing the number of ED visits in the CG (n = 24; 30%) and the IG (n = 4; 4.9%), we observed significant differences with a medium-high effect size in favour of the group that participated in the nurse-led telecare follow-up programme (U = 2347.5; *p* < 0.001; η^2^ = 0.133). However, no significant differences were observed between the CG (n = 3; 3.8%) and the IG (n = 4; 4.9%) in terms of the number of patients requiring readmission after emergency department visits (U = 3238.5; *p* = 0.988). Similarly, there were no significant differences in the number of patients requiring re-intervention after ED visits when comparing the CG (n = 1; 1.2%) and the IG (n = 0; 0%) (U = 3200.0; *p* = 0.320). IG patients who visited the ED and were readmitted had significantly shorter hospital stays than those in the control group (IG = 6 ± 1.41 days vs. CG = 23.33 ± 17.39 days) with a very high effect size (U = 0.000; *p* < 0.05; η^2^ = 0.654) (Table 4).

## 4. Discussion

Our study highlights the importance of the implementation of a post-surgical telecare follow-up programme led by a nurse trained in bariatric surgery. It evaluated the number and type of calls received in the immediate postoperative period from patients who had various different issues of concern following surgery. The results revealed a significant reduction in the number of visits that these patients made to the emergency department.

The implementation of telemedicine technologies has transformed patient postoperative follow-up. Several studies have been published on patients who have undergone bariatric surgery where an easily accessible telecare follow-up programme was carried out in a similar way [43,44]. Neil et al. [43] highlight the importance of nurse expertise in the perioperative period for patients undergoing bariatric surgery, with the aim of providing sensitive and high-quality care to these patients. They argue that nurses are the key point of contact and the main care provider during all perioperative phases for this type of patient. Dolne et al. [20] investigated nursing performance when caring for bariatric surgery patients, emphasising the need for close observation and careful monitoring to detect any signs of potential complications.

Ruiz-Tovar et al. [16] clearly emphasise the importance of nurses within the multimodal rehabilitation programme for patients who have been discharged following bariatric surgery. When measures are implemented, they must be tailored to the organisational structure of each institution and ensure that nursing is an essential part of the process. In the study by Arnaert et al. [44], the participants welcomed the idea of integrating telenursing into care programmes following bariatric surgery, as they saw this new approach to care as a way of overcoming the current challenges in accessing bariatric services. The participants highlighted that the most significant benefit was the timely advice and care provided by the nurse. This personalised approach to nursing, which gave patients direct access to a frontline professional, empowered the participants to exercise greater control over their recovery process, promoting self-management and enhancing feelings of confidence and peace of mind.

To date, different studies have been conducted with surgical patients, incorporating telephone calls in the postoperative period with the aim of providing closer support to these patients at a time that is fraught with uncertainty and fear. The randomised pilot study of colorectal cancer patients carried out by Harrison et al. [45] found promising indications for health system outcomes that warranted further study and hospital readmissions. Shah et al. [46] evaluated the impact of telephone calls within 72 h of discharge in patients following head and neck surgery, showing statistically significant reductions in emergency department visits. Unlike these studies, in which calls were made directly to the patient on specific days, the patients in our study had the freedom to contact the nurse whenever they had a problem, query, or concern. We believe that this approach, in which patients decide when to make contact, gives the patient more autonomy and control in the process of managing their issues. De Dicastillo et al. [47] developed and evaluated a telematic platform for monitoring patients in outpatient surgery, highlighting its effectiveness in reducing complications and improving patient satisfaction.

The main results of our study showed a significant number of consultations made (4.3 consultations per patient on average), primarily via WhatsApp, which is indicative of the level of concern among patients who have recently undergone bariatric surgery. Concurring with previously published research [48], the main reason why the bariatric patients participating in our study contacted the on-call nurse was to communicate diet-related concerns. This finding suggests that it could be important to make sure that nutritional surveillance interventions are incorporated into the post-surgical care offered to bariatric patients [49]. The second-most-frequent concern was related to drainage problems (mainly discomfort), which raises the question of whether or not it is really necessary to discharge all patients with an abdominal drainage, especially given the controversy around this topic [50,51]. Since being discharged with an abdominal drainage is a main cause for concern amongst patients who have undergone bariatric surgery and some research suggests that there could be no benefit in discharging bariatric patients with an abdominal drainage [50,51], surgeons should reflect on this practice and make decisions based on each individuals’ needs. The next most common problems raised by the patients were in relation to surgical wounds, abdominal pain, intestinal transit disorders, and vomiting, which have been highlighted as the main causes and risk factors related to ED visits, as well as 30- and 90-day readmissions in patients who have undergone laparoscopic gastric bypass and laparoscopic vertical gastrectomy [28].

Our results highlight the benefits of a telecare follow-up programme, given that the number of visits to the emergency department in the IG was significantly lower than in the CG (24 vs. 3; *p* < 0.01). These findings are in line with those presented by Kenawy et al. [52], whose study concluded that calling and talking to a patient following bariatric surgery was directly related to a 52% decrease in non-emergency hospital visits 90 days after bariatric surgery. This system could reduce costs for both the patients and the hospital.

In summary, our results reflect trends and practices observed in the existing literature on bariatric surgery, highlighting the importance of adequate postoperative follow-up and the use of communication technologies to improve patient outcomes.

This study has several limitations that need to be highlighted. Firstly, using a quasi-experimental design instead of a controlled randomised trial and recruiting patients from a small geographical area limit the generalizability of the results. Secondly, at the time when the study was designed, no evidence had been found on the effects of nurse-led telecare interventions on the number of ED visits in the first 30 days after bariatric surgery; therefore, the study sample needed had to be calculated based on an approximation and this could have affected the power of the study. Thirdly, data collection was limited exclusively to readmissions in the hospitals included in the study. We cannot ascertain whether the participants were admitted to other hospitals.

## 5. Conclusions

In conclusion, our study reveals that the postoperative care of patients who have undergone bariatric surgery requires a multidisciplinary approach that combines clinical assessment, nursing interventions, and the use of telematic technologies in order to best respond to patient needs. The implementation of telemedicine and continuous post-discharge communication led by a bariatric nurse specialist has had highly beneficial results in resolving problems at discharge and reducing hospital readmissions, thus improving patient quality of life.

Future research should focus on the promotion and optimisation of these practices to provide more effective and personalised care. In addition, it would be beneficial to further explore the impact of different nursing intervention strategies in diverse clinical and cultural contexts, as well as to evaluate the sustainability and cost-effectiveness of telematic technologies in postoperative care. It is essential to provide nurses with ongoing training in the use of these technologies and the implementation of evidence-based protocols in order to maximise the benefits of these innovations in healthcare. Lastly, it is crucial for physicians, nurses, and other healthcare professionals to collaborate to ensure comprehensive and effective care for patients who have undergone bariatric surgery.

## Figures and Tables

**Table 1 healthcare-12-02448-t001:** Summary of sociodemographic and clinical data.

Characteristics	All Participants (*n* = 161)	Control Group (*n* = 80)	Intervention Group (*n* = 81)	*p*-Value
Gender (*n*, %)				
Female	106 (65.8%)	57 (71.3%)	49 (60.5%)	0.150
Male	55 (34.2%)	23 (28.7%)	32 (39.5%)	
Age (years)	44.21 ± 9.65	45.24 ± 8.63	43.20 ± 10.52	0.122
BMI (Kg/m^2^)	44.52 ± 7.35	46.25 ± 6.69	42.81 ± 7.60	0.062
Comorbidities (*n*, %)				
Hypertension	62 (38.5%)	32 (40.0%)	30 (37.0%)	0.700
Diabetes	27 (16.8%)	17 (21.3%)	10 (12.3%)	0.132
Obstructive sleep apnoea	61 (37.9%)	22 (27.5%)	39 (48.1%)	0.008
Other	81 (50.3%)	26 (32.5%)	55 (67.9%)	<0.001

**Table 2 healthcare-12-02448-t002:** Timing of telephone consultations.

Time Frame	Frequency	%
Morning (07:00–15:00)	194	54.3
Afternoon/evening (15:00–23:00)	150	42.0
Night (23:00–07:00)	13	3.6
Total	357	100

**Table 3 healthcare-12-02448-t003:** Reasons for consultation.

		Frequency	%
Reason	Diet concerns	93	26.1
	Protocol concerns	1	0.3
	Drainage problems	40	11.2
	Fever	6	1.7
	Abdominal pain	17	4.8
	Vomiting	13	3.6
	Wounds	20	5.6
	Constipation	17	4.8
	Pyrosis	9	2.5
	Dizziness	3	0.8
	Diarrhoea	7	2.0
	Other	131	36.7
Total		357	100

**Table 4 healthcare-12-02448-t004:** Between-groups comparison: ED visits and readmissions.

Characteristics	Control Group (N = 80)	Intervention Group (N = 81)	*p*
Visit to ED (n, %)	24 (30.0%)	4 (4.9%)	<0.001
Reason for ED visit			
*Drainage problems*	7 (8.8%)		
*Fever*	2 (2.5%)		
*Abdominal pain*	3 (3.8%)	3 (3.7%)	
*Vomiting*	3 (3.8%)		
*Wound concerns*	5 (6.4%)		
*Constipation*	1 (1.3%)		
*Dizziness*	3 (3.8%)		
*Other*		1 (1.2%)	
Readmissions	3 (3.8%)	4 (4.9%)	0.988
Surgical re-intervention	0 (0%)	1 (1.2%)	0.032
Mean length of readmission stay (in days)	23.33 ± 17.39	6 ± 1.41	<0.05

## Data Availability

The raw data supporting the conclusions of this article will be made available by the authors on request.

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
