# Peer review of "Effectiveness of a Nurse-Led Telecare Programme in the Postoperative Follow-Up of Bariatric Surgery Patients: A Quasi-Experimental Study"

_healthcare, 2024, doi:10.3390/healthcare12232448_

Round 1
Reviewer 1 Report
Comments and Suggestions for Authors
Dear authors,
I enjoyed reading the manuscript and I congratulate you on the importance of the topic they chose. With a view to improvement, I present below some notes.
Title. The title is clear and concise. It gives an overall idea of the study.
Abstract. Allows readers to give an overall idea of the study.
Keywords. To increase the possibility of citation of the manuscript, the use of MesH descriptors is suggested.
Introduction. Some research on nurse-led multidisciplinary interventions needs to be described so that readers know the state of the art. Some of these interventions use e-health technologies alone and in combination with face-to-face consultations. Also, provide readers with a definition of the concept of telecare. Explain globally how surgical patients can be followed up after discharge using telecare. One of Trend's criteria is “Theories used in designing behavioural interventions”. What nursing theories support the development of multidisciplinary telecare interventions? I suggest that the authors consult the article by Nagel and Penner (2015), entitled “Conceptualizing Telehealth in Nursing Practice: “Advancing a Conceptual Model to Fill a Virtual Gap” (DOI: 10.1177/0898010115580236).
Materials and methods. How was the sample size calculated? Was any software used?
Line 110. What sampling method was used?
Line 123. How were the patients selected for the control group and the intervention group? How were the patients chosen for the control group and the intervention group? In the abstract, the authors say that the study was carried out in two hospitals, but this information is not in this section.
Lines 127-151. The intervention is limited solely to the possibility for patients to consult a specialized nurse over the phone. However, the title gives readers the idea that this is a multi-intervention program. Please clarify.
Lines 154-194. Of the set of variable outcomes, only a few were used to evaluate the effect of the intervention, while the rest were used to characterize the patients. Please organize the variables in such a way as to let the readers know which variables were used to characterize the patients and which were used to evaluate the effect of the intervention.
Lines 224-225: please include the citation of the classification method used.
Please include a subsection for ethical considerations. Indicate whether informed consent was sought and whether participation was voluntary.
Results. Line 300. Indicate the number of consultations per patient on average.
Lines 302-314. These sentences repeat the information described between the lines 236-239. Instead of repeating the information in this section, please discuss these results in the light of the literature.
Author Response
Please, see the attachment.

Reviewer 2 Report
Comments and Suggestions for Authors
This is a quasi-experimental study. The research design has some flaws. It's not a randomized trial.
The primary endpoint of the study is the number of visits to the emergency department. The author said, "It is also apparent that other programmes with a post-surgery call from healthcare staff reduce ED visits by 54% in the first 90 days". So the sample size calculation is based on that. It's unreasonable and it's not scientific. The results are different between 90 days and 30 days of follow-up.
Is there a nurse for the intervention? Is it the same nurse for intervention and recruitment?
How do you ensure the quality of the data? Do you have a double check?
The outcome variables section is too confusing.
The discussion section needs to be revised. The discussion is not a repetitive description of the results.
How do you explain the difference in readmissions and surgical re-intervention between 2 groups?
Comments on the Quality of English LanguageThe English could be improved to more clearly express the research.
Author Response
Please, see the attachment.

Reviewer 3 Report
Comments and Suggestions for Authors
I thank the authors of the manuscript "Effectiveness of a Nurse Telecare Programme in the Postoperative Follow-up of Bariatric Surgery Patients" for the time they have devoted to conducting their scientific work and for the time they have devoted to writing their manuscript. The manuscript appears to be of interest, and is currently of great interest to the scientific community. The objectives of this study were: (a) to describe the nature of consultations in patients who have undergone bariatric surgery in the immediate postoperative period after discharge (first month post-surgery) and (b) to analyse the effects of the intervention (implementation of a nurse-led telecare post-surgical follow-up programme) on the complications and readmissions of these patients. The hypothesis of this study was that a nurse-led telecare follow-up programme reduces complications and optimises the safety and quality of immediate post-surgical care in people who have undergone bariatric surgery. The introduction of the manuscript was written in a precise manner and aimed at introducing the subject matter and purpose of the study. The materials and methods section adequately describes the methodology used to conduct the study. The results reflect the data obtained by the authors in conducting their study. The discussion was clearly written. Conclusions are relevant to the results obtained. I advise authors to highlight if there are statistically significant differences in the condition shown in Table 1, that is, between the control group and the intervention group there are higher mean ages and BMIs in the control group. I would specify in the text what are the other co-morbidities (n. 81; 50.3%) reported in Table 1. I recommend to the authors to report more details regarding the no. 131 reasons of consultation reported in Table 3.
Author Response
Please, see the attachment.

Round 2
Reviewer 2 Report
Comments and Suggestions for Authors
None
Author Response
Thank you for your comments.